# Human response times are governed by dual anticipatory processes with distinct neural signatures

Ashwin G. Ramayya [1] ✉, Vivek Buch [1], Andrew Richardson [2], Timothy Lucas[3] & Joshua I. Gold [4]

Human behavior is strongly influenced by anticipation, but the underlying neural mechanisms are poorly understood. We obtained intracranial electrocephalography (iEEG) measurements in neurosurgical patients as they performed a simple sensory-motor task with variable (short or long) foreperiod delays that affected anticipation of the cue to respond. Participants showed two forms of anticipatory response biases, distinguished by more premature false alarms (FAs) or faster response times (RTs) on long-delay trials. These biases had distinct neural signatures in prestimulus neural activity modulations that were distributed and intermixed across the brain: the FA bias was most evident in preparatory motor activity immediately prior to response-cue presentation, whereas the RT bias was most evident in visuospatial activity at the beginning of the foreperiod. These results suggest that human anticipatory behavior emerges from a combination of motor-preparatory and attention-like modulations of neural activity, implemented by anatomically widespread and intermixed, but functionally identifiable, brain networks.

Expectations of future events shape human behavior[1,2]. Even simple stimulus detection is not a passive, purely stimulus-driven process but involves predictive inference that combines incoming sensory information with expectations learned from prior experience[3,4]. However, the mechanisms in the human brain that allow expectations to influence impending sensory-motor processing (a set of phenomena that we refer to as "anticipation") remains unclear.

Anticipatory influences on human behavior have been quantified extensively using simple sensory-motor detection tasks[5,6]. For example, the variable foreperiod-delay paradigm has been used to operationalize anticipatory behavior in terms of changes in response times (RTs) and premature responses ("false alarms", or FAs) as a function of changes in the length and predictability of the foreperiod delay[6-8]. These effects are thought to reflect modulations of preparatory motor processes during the foreperiod delay[2,9-11]. These preparatory motor processes are often modeled via "rise-to-bound" dynamics that account for endogenous RT variability as arising from a stochastically varying processes that triggers a motor response[12-18]. In these models, anticipatory response biases are often assumed to arise from prestimulus elevations in the baseline, or "starting point," of the stochastic rising process, resulting in faster RTs[13,14]. A compelling feature of these models is that, in addition to providing parsimonious accounts of behavior, they have algorithmic components that are thought to map directly onto the

activity patterns of localized sets of neurons that contribute to motor preparation and execution[19].

However, exactly how these models relate to sensory-motor processing in the human brain is not well understood, reflecting a lack of brain measurements with appropriate combinations of high spatiotemporal resolution and broad anatomical scale. Scalp electroencephalography (EEG) studies have provided support for anticipatory processing during the foreperiod delay[19-23], but it is difficult to interpret these signals in terms of specific neural circuits because they aggregate activity across large brain regions. Functional MRI studies have shown regionally distributed hemodynamic correlates of anticipatory processing, but these findings have been inconsistent and are difficult to relate to RT variability because of limited temporal resolution[24-26].

To overcome these limitations, we obtained high-resolution intracranial electroencephalography (iEEG recordings) from 23 patients with medically refractory epilepsy with indwelling intraparenchymal electrodes in widespread brain regions as they performed a stimulus-detection task with a variable foreperiod delay (Fig. 1A, Table S2). We focused on high-frequency iEEG activity, which reflects local spiking activity (70–200 Hz power) sampled broadly across many parts of cortex and certain subcortical structures[27-30]. The central hypothesis that motivated and guided this study was that neural populations in the human brain that

[1]Department of Neurosurgery, Stanford University, Stanford, CA, USA. [2]Department of Neurosurgery, Hospital of University of Pennsylvania, Philadelphia, PA, USA. [3]NeuroTech Institute, Columbus, OH, USA. [4]Department of Neuroscience, University of Pennsylvania, Philadelphia, PA, USA. ✉e-mail: aramayya@stanford.edu

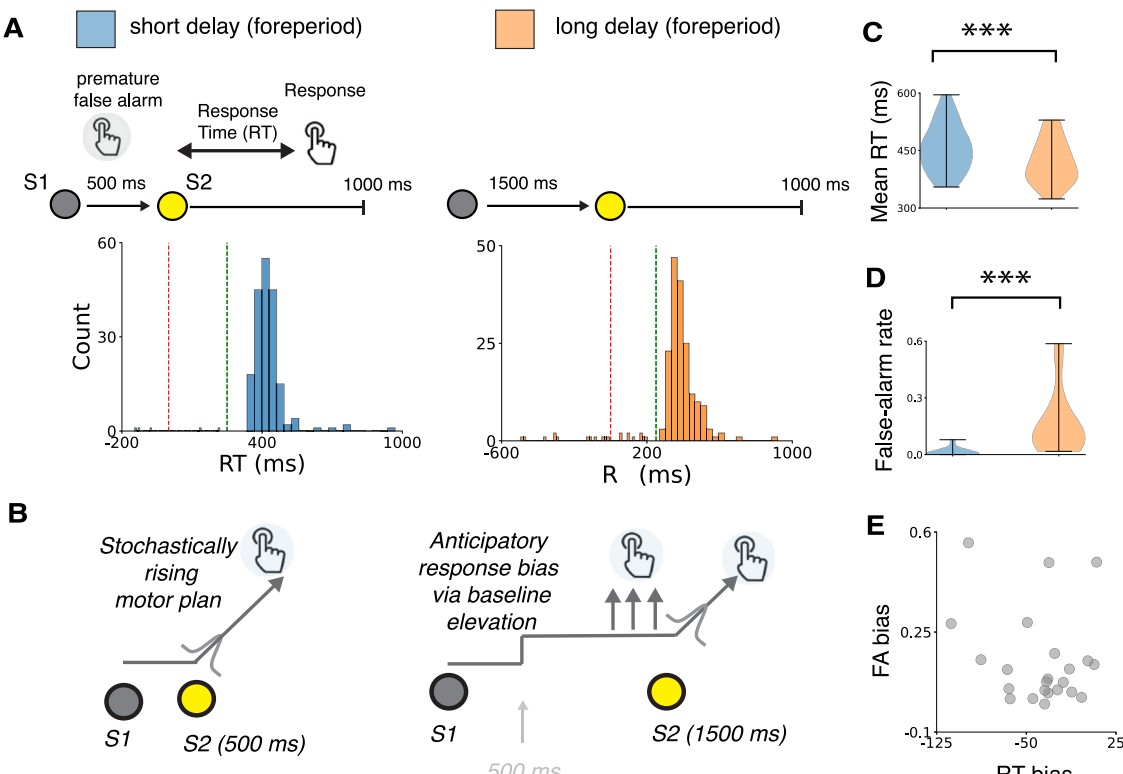

**Fig. 1 | Two forms of anticipatory biases. A** Task summary and RT distributions from an example participant. Red line indicates stimulus onset, green vertical line indicates 250 ms after stimulus onset (the fast-response threshold). Blue and orange histograms indicate timing of responses on short- and long-delay trials, respectively. Premature false alarms are responses that fall to the left of the red line. **B** Model schematic illustrating abstracted preparatory motor processes contributing to sensory-motor behavior. Anticipatory elevation of baseline activity can account for both a decrease in RT and an increase in false-alarm rate. Violin plots showing distributions of mean RTs (**C**) and premature false-alarm rates (**D**) on short- (blue) and long- (orange) delay trials for all 23 participants. **E** Scatterplot showing covariance of delay-related changes in mean RT and premature false-alarm rate across participants. Each circle corresponds to data from a single participant.

support motor preparation (and thus would tend to exhibit activity patterns that vary systematically with RT) encode anticipatory biases via modulations of prestimulus activity. We focused on a broad set anatomical regions because neural correlates of RT variability have been identified in several motor-preparatory brain regions, including activity patterns that map directly[19,31] or indirectly[32–34] rise-to-bound model dynamics. As detailed below, our results identify two behaviorally and neurally distinct processes that govern anticipatory effects on sensory-motor behavior and highlight the complex but identifiable mappings between algorithm- and implementation-level explanations of human behavior.

## Results

Twenty-three participants performed a variant of a commonly used "foreperiod-delay task" that has been used extensively to investigate anticipatory influences on sensory-motor behaviors[6,7,35]. Briefly, each trial began with the presentation of a visual target ("warning signal," S1) on a computer screen that changed color after a randomly selected foreperiod delay of 500 ("short") or 1500 ("long") ms. Participants were instructed to respond via button press as soon as they noticed the color change ("stimulus," S2). RT was measured as the elapsed time between stimulus and response. The different foreperiod delays provided categorically different levels of temporal expectation of stimulus arrival at the time of stimulus presentation[2,6,8,11]. On short-delay trials, the stimulus was presented when there was uncertainty about whether the trial was a short- or long-delay trial, resulting in relatively low temporal expectation of stimulus arrival. On long-delay trials, the stimulus was presented when the trial could be identified unequivocally as a long-delay trial, resulting in relatively high temporal expectation of stimulus arrival (Fig. 1).

### Dual behavioral signatures of anticipatory biases

The participants' RTs included endogenous variability for both delay conditions (median per-participant RT inter-quartile range = 66.67 ms for short-delay and 66.75 for long-delay trials; examples are shown in Fig. 1A, Table S3), with two primary effects of anticipation that were consistent with previous findings[6,8,14,36]. First, participants had faster RTs (paired $t$-test, $t$ (22) = 5.57, $p < 0.001$) on long- versus short-delay trials ("RT bias"), albeit with substantial individual variability (mean RT range across participants = 354–595 ms and 323–529 ms on short- and long-delay trials, respectively, Fig. 1C). Second, participants had higher false-alarm rates on long- versus short-delay trials ("FA bias"; paired $t$-test, $t$ (22) = 4.43, $p < 0.001$; range 1.7–58%, versus 0–7.8%, respectively, Fig. 1D). We did not observe a significant correlation with delay-related differences in RT and false-alarm rate ($p > 0.3$, Fig. 1E) but did observe a correlation between RT and false-alarm rate when considering only long-delay trials ($r = 0.47$, $p = 0.02$).

We modeled these anticipatory effects as prestimulus modulations of an abstracted "rise-to-bound" motor-preparatory process (Fig. 1B)[14]. Specifically, we modeled each RT on short-delay trials as the time taken for a latent variable to rise from a fixed starting point to a fixed bound value to trigger a motor response ("rising process"). Trial-to-trial variability in the rate of rise accounts for endogenous RT variability and the characteristic (delay-independent) right-tailed RT distribution. For correct trials with RT > 250 ms, we assumed that this rising process was triggered by the onset of the stimulus S2. In contrast, for trials with false alarms, we assumed that this rising process was triggered prior to the onset of S2, according to a stochastic process that occurred with uniform probability during the 500 ms preceding S2. We modeled each RT on long-delay trials as emerging from a rising process that was modulated by increased temporal anticipation. This anticipation took the form of an elevation of the baseline starting point of the

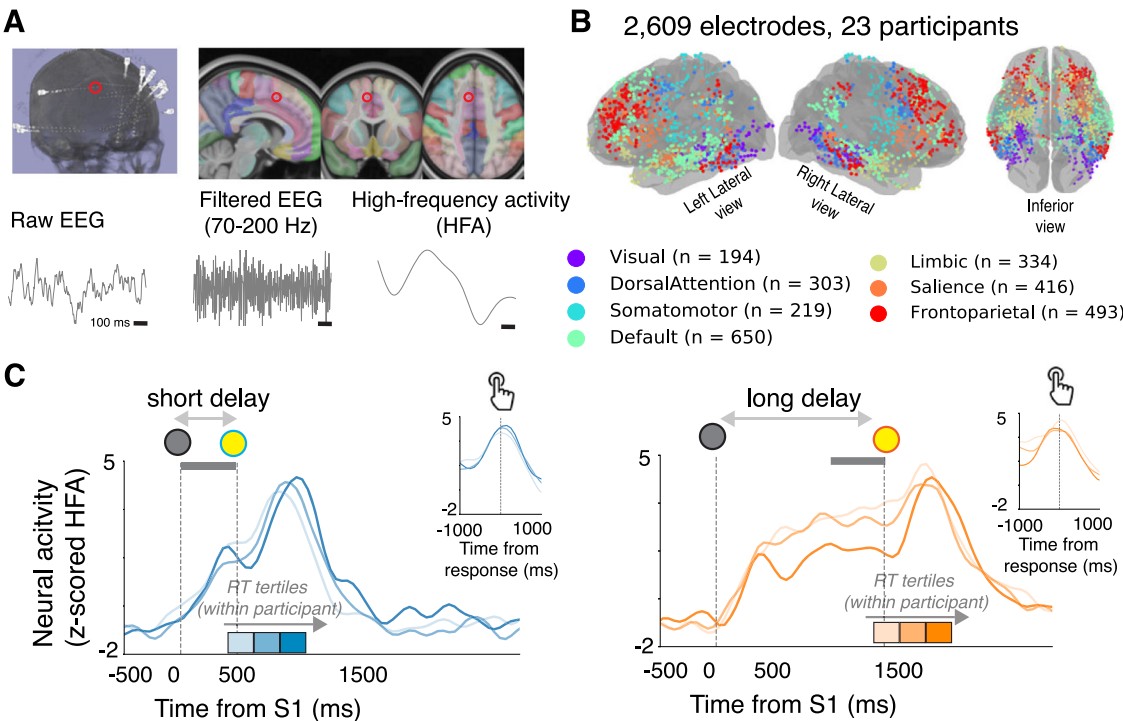

**Fig. 2 | Intracranial recordings provide neural measurements with high resolution and broad anatomical coverage. A** Extracting high-frequency activity (HFA, 70–200 Hz power) from intraparenchymal depth electrodes as an estimate of local spiking (red circle indicates example prefrontal electrode described in **C**). **B** Brain plot showing electrode locations from all participants in standard MNI coordinates. Colors indicate intrinsic brain networks based on a normative atlas (Yeo et al., 2011). **C** Task-driven responses of local neural activity (average z-scored HFA) measured at the electrode indicated in (**A**), plotted separately for short- (left panel, blue) and long- (right panel, orange) delay trials and binned by RT tertiles (3 bins, computed within each participant; lighter shading indicates faster RTs). Main panels shows target-locked activity. Vertical lines indicate time of target onset (gray), short-delay color change (blue), and long-delay color change (orange). Insets on top right show response-locked activity. Shaded gray box indicates the time interval during which we observed a correlation between neural activity and RT variability.

rising process, bringing the process closer to the threshold value required to trigger a response. This baseline elevation increased the probability of triggering responses prior to S2 (FA bias) and reduced the time taken to generate a correct response after S2 (RT bias). We used separate parameters for anticipatory baseline elevation to trigger premature responses versus speed up correct RTs, to account for largely independent variability in RT and FA biases across participants.

This anticipatory starting-point model provided a good fit to participants' RTs and anticipatory response biases ($R^2$ mean = 0.85, range = 0.62–0.97; Fig. S1 for individual model fits). We compared these fits to the fits of two additional models that included alternative mechanisms to explain anticipatory RT biases (the three models had the same number of free parameters such that it was not necessary to apply a penalty for model complexity). The first alternative model replaced adjustments to the starting point with adjustments to the variance of the rate-of-rise distribution. This "variance model" generated poorer fits ($R^2$ mean = 0.77, range = 0.37–0.97) than the starting-point-only model. The second alternative model replaced adjustments to the starting point with adjustments to the mean of the rate-of-rise distribution. This "mean model" produced similar fits ($R^2$ mean = 0.86, range = 0.62–0.97) as the starting-point-only model. We drew similar conclusions by comparing model simulations to group-level behavioral data (Fig. S2). These results, particularly the similar fits of the starting-point-only and mean models, highlight the difficulty in uncovering mechanisms of anticipatory behavior using algorithmic modeling alone[6,11]. Below we focus on the more parsimonious starting-point-only model to identify the underlying neural mechanisms.

**Task- and RT-modulated neural responses were distributed widely in the brain**
We obtained neural measurements from intraparenchymal depth electrodes implanted in participants with medically refractory epilepsy for clinical purposes (Fig. 2A). We focused on high-frequency activity (HFA, 70–200 Hz power) recorded from bipolar pairs of electrodes, which provides a reliable surrogate of local (within ~3 mm) neural population spiking activity[27–29]. In total, we studied recordings from 2609 bipolar pairs of intraparenchymal depth electrodes distributed widely throughout the brain in 23 patients (mean = 113.4 electrodes/participant). We localized these recordings to various intrinsic brain networks (Fig. 2B). For each electrode, we measured task-related activity of the nearby neural population time-locked to target onset and motor response in ~50 ms sliding time intervals, z-scored to an aggregate baseline from the entire recording session.

We identified delay-related modulations and/or correlations with delay-independent RT variability at various time intervals throughout the trial, after accounting for transient sensory- and motor-driven responses, from 2142 out of 2609 electrodes (Fig. S3). To identify delay-related changes in activity, we compared neural activity following the warning signal (500 ms following S1), stimulus onset (500 ms following S2), and response onset (1000 ms following the button press), relative to a baseline interval (500 ms prior to S1; paired t-tests, $p < 0.05$). To relate neural activity at each electrode with delay-independent RT variability, we used a multivariate model that included neural activity in various task-related time intervals (significance via non-parametric shuffle procedure $p < 0.05$; Fig. S3).

An example electrode showing both delay- and RT-related activity modulations is shown in Fig. 2C (the electrode location is indicated in Fig. 2A). This local neural population showed rising activity following the warning signal (S1) that peaked near the time of response and descended back to baseline. The prestimulus baseline activity was relatively higher during faster RT and long-delay trials, followed by a largely RT-independent rate of rise before the motor response. In other words, this electrode showed modulations that were roughly consistent with "rise-to-bound" dynamics found in our and similar models.

**Fig. 3 | Task-related neural activity changes were regionally widespread and intermixed.**
**A** Percentages of electrodes in each region with activity that showed task-related activity changes following S1, S2, or R. Positive values indicate task-related increases in activity; negative values indicate task-related decreases in activity. **B** Scatterplot showing the relative frequency of electrodes with positive (ordinate) and negative (abscissa) changes in activity to task-relevant events in each intrinsic brain network relative to their overall (expected) frequency across the brain (z-scores). Positive values indicate increased relative frequency; negative values indicate decreased relative frequency. Inner and outer ellipses indicate 1σ and 2σ confidence intervals derived from the joint distribution, respectively. **C, D** Same as (**A**, **B**), but for positive and negative correlations with trial-to-trial, delay-independent RT variability during any time interval. Negative values in (**C**) indicate increased activity with faster RTs; positive values indicate increased activity with slower RTs.

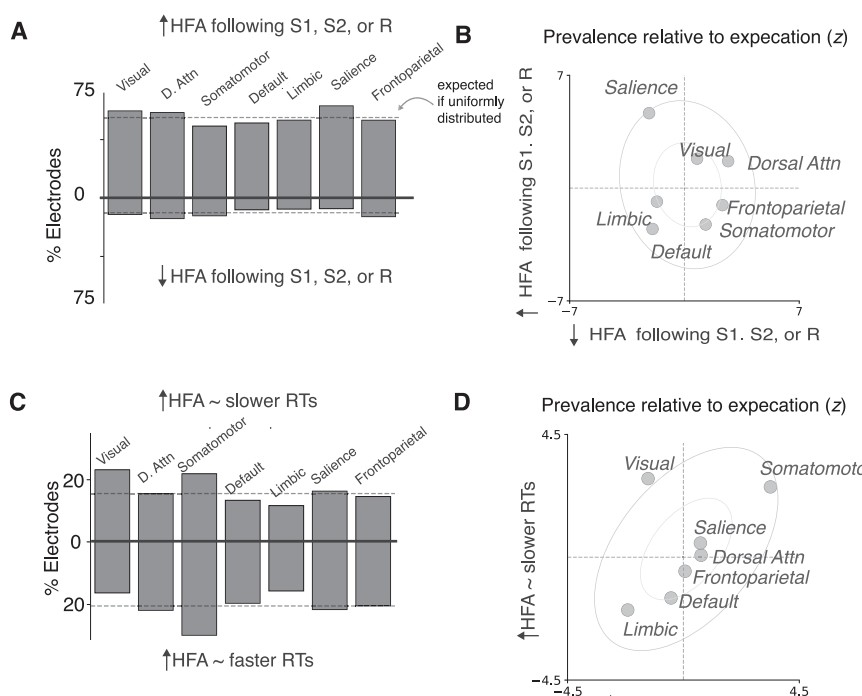

However, this electrode was hardly unique in showing task and RT modulations, which we found throughout the brain (Fig. 3). Task modulations across all task epochs typically involved activity increases, rather than decreases, in response to task-relevant sensory and/or motor events (any interval; two-sided binomial test $p < 0.001$, 95% CI: 83–86%; expected 50%), and rarely included both task-driven increases and decreases in different task epochs measured at the same electrode ($n = 65/2,609$, 3%). Task-related increases were generally distributed uniformly across the brain, but neural populations in the salience network showed more frequent task-driven increases than expected (two-tailed binomial test, FDR-corrected $p < 0.001$, 95% CI: 74–85%; Fig. 3).

RT modulations were also distributed widely, but with different spatial patterns. We observed a regional intermixing of effects that included both positive correlations, such that increased activity corresponded to slower RTs, and negative correlations, such that decreased activity correlated with faster RTs. Neural activity showed more frequent correlations with RT than chance across the brain ($n = 638/2609$, one-tailed binomial test $p < 0.001$, 95% CI > 23%, where chance=5%) and within each intrinsic brain network (FDR-corrected $p$s < 0.001, 95% CIs >15–30%), even when separately considering only positive or negative correlations (corrected $p < 0.03$, except limbic populations rarely showed positive correlations, corrected $p > 0.5$). We rarely observed neural populations at a single electrode that showed both positive and negative RT correlations during different task epochs ($n = 60/2,609$, 2%). These RT modulations were not distributed uniformly across the brain: visual neural populations showed positive RT correlations more frequently than expected (two-tailed binomial test, corrected $p = 0.04$; 17–30%), whereas somatomotor neural populations showed negative RT correlations more frequently than expected (corrected $p = 0.02$; 24–37%). We observed similarly distributed and intermixed neural representations when using finer-grained anatomical parcellations (Fig. S4).

**Data-driven clustering of neural activity patterns**
To better understand these broadly distributed, diverse activity patterns, we used a data-driven hierarchical clustering algorithm to group electrodes that showed similarities in delay-related activity modulations and delay-independent RT correlations (Fig. 4). We selected a clustering level (4) that maximized the number of clusters that that included data from all of the participants (we also excluded clusters with <200 electrodes) and exhibited

distinct patterns of modulations by task events and RT. Cluster 0 showed task-related increases without RT modulation. Cluster 1 showed task-related increases with negative RT correlations. Cluster 2 showed task-related decreases without RT modulation. Cluster 3 showed task-related increases with positive RT correlations. These clusters were distributed widely across the brain (see Fig. S3).

**Dual neural signatures of anticipatory biases in prestimulus activity**
Prestimulus activity in two distinct neural clusters encoded participant-to-participant variability in the two main anticipatory biases we identified from behavior: RT bias in Cluster 3, and FA bias in Cluster 1 (Fig. 5). Specifically, we measured prestimulus activity in each cluster on long-delay trials (ranging from 250 ms prior to S1 to 50 ms prior to S2, excluding trials with FA and RTs <250 ms, averaged within participants) and related these participant-wise measures to anticipatory RT and FA biases (as shown in Fig. 1E). We found that increased prestimulus activity in Cluster 1 correlated with increased FA bias ($\rho = 0.53$, corrected $p = 0.04$; partial correlation controlling for RT bias). In contrast, increased prestimulus activity in Cluster 3 correlated with RT bias ($\rho = 0.62$, corrected $p = 0.002$). We further detail the nature of prestimulus modulations in these two different clusters below.

The two clusters associated with anticipatory biases exhibited different temporal neural dynamics (Fig. 6). We performed a time-resolved partial-correlation analysis relating prestimulus activity in Cluster 1 with FA bias (Fig. 6A; 250 ms sliding advanced 10 ms steps, controlling for RT bias) and prestimulus activity in Cluster 2 with RT bias (Fig. 6D; controlling for FA bias). Cluster 1 showed peak correlation strength immediately preceding S2 (1180–1430 ms following S1, highlighted in Fig. 6B; a scatterplot of this correlation is shown in Fig. 6C), whereas Cluster 3 showed peak correlation strength around the time of S1 onset (120 ms prior to and 30 ms following S1, highlighted in Fig. 6E; a scatterplot of this correlation is shown in Fig. 6F). These temporal dynamics were evident in the time course of neural activity during long-delay trials in participants grouped by the degree to which they showed anticipatory biases (Fig. 6B, E). Participants who showed stronger FA biases showed increased ramping activity as the time of S2 drew nearer (Fig. 6B), as expectation of stimulus arrival is presumably increasing (as illustrated by our modeling analysis; Fig. 1B). Participants who showed stronger RT biases showed stronger transient increases in activity after the

**Fig. 4 | Data-driven clustering of task-related neural activity.** Illustration of hierarchical clustering of electrodes based on similar task-related activity modulations. Top row: time intervals used to measure task modulation and RT correlation for each electrode as detailed in main text. Bottom row: left, colormap representing a feature matrix across all electrodes, where each row represents an electrode, and each column represents a feature (task or RT modulation); middle: dendrogram representing similarities between electrodes; right: feature matrix re-organized based on similar task and RT modulations at the clustering level indicated by the black line in middle panel, which we used in these analyses. **B** We identified a level of clustering (black vertical line in dendrogram) based on an objective function that maximized the number of clusters that were well sampled in our dataset (i.e., each eligible cluster was observed in all participants and consisted of at least 200 electrodes). **C** Percentage of electrodes that showed task-related increases and decreases (positive and values, respectively as in Fig. 3A). **D** Same as (**C**) but for RT correlations. Negative values indicate increased activity with faster RTs; positive values indicate increased activity with slower RTs (as in Fig. 3C).

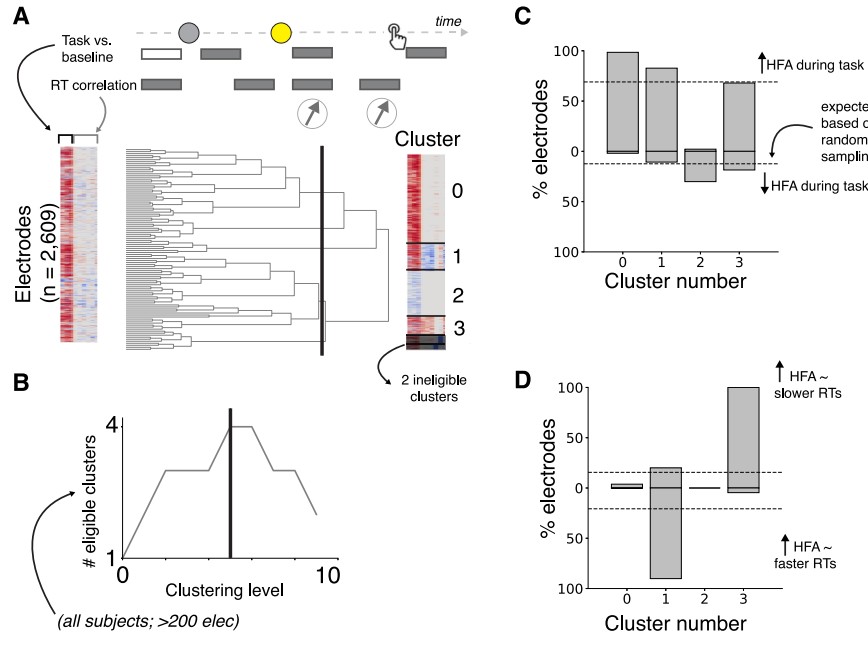

activity corresponded to faster RTs (Fig. 7A, B; Fig. S5). Cluster 3 neural activity patterns that encoded RT biases were activated more strongly by sensory than motor events (Fig. 7D–F). These activity patterns, on average, tended to show a transient rise in activity after the onset of the warning signal and did not show rising activity preceding response, including on FA trials (Fig. 7F). They showed positive correlations with endogenous RT variability such that increased activity (including prior to S1) corresponded to slower RTs (Fig. 7D, E; Fig. S5).

These results suggest a functional heterogeneity in prestimulus activity related to anticipatory processing, such that Cluster 1 activity prior to S2 encodes FA bias, whereas Cluster 3 activity around the time of S1 encodes RT bias. We tested for such a functional dissociation using a Linear Mixed Model (LMM) analysis, in which we related prestimulus activity (z-scored HFA) to RT bias (interacting with "cluster," and "time window," defined based on peak correlations from the previous analysis) and FA bias (also interacting with "cluster" and "time window"), with "participant" and "intrinsic brain network" as random effects (varying intercepts; see Materials and Methods). We observed a significant three-way interaction between RT bias, cluster, and time window ($F = 3.93$, $p = 0.005$), and between FA bias, cluster, and time window ($F = 19.69$, $p < 0.001$), suggesting a functional heterogeneity in prestimulus activity modulations during anticipatory processing.

Cluster 1 and 3 electrodes were widely distributed and regionally intermixed across the brain, but each showed non-uniform distributions (Fig. 8A–C). Cluster 1 electrodes were found more frequently in somato-motor networks (two-tailed binomial test, corrected $p = 0.02$; found 15–27%, expected 14%), and less frequently in visual and limbic networks (corrected $p$s < 0.03; 03–13%), than expected. Cluster 3 electrodes were found more frequently in visual networks (two-tailed binomial test, corrected $p = 0.02$; found 12–23%, expected 10%) than expected. The visual network was an outlier in showing an increased probability of containing Cluster 3 electrodes, and a decreased probability of containing Cluster 1 electrodes (Fig. 8C).

## Discussion

We identified neural correlates in the human brain of two behaviorally distinguishable effects of anticipation on a simple sensory-motor behavior. The first effect, which we refer to as a false-alarm (FA) bias, was characterized behaviorally by an increase in premature responses under

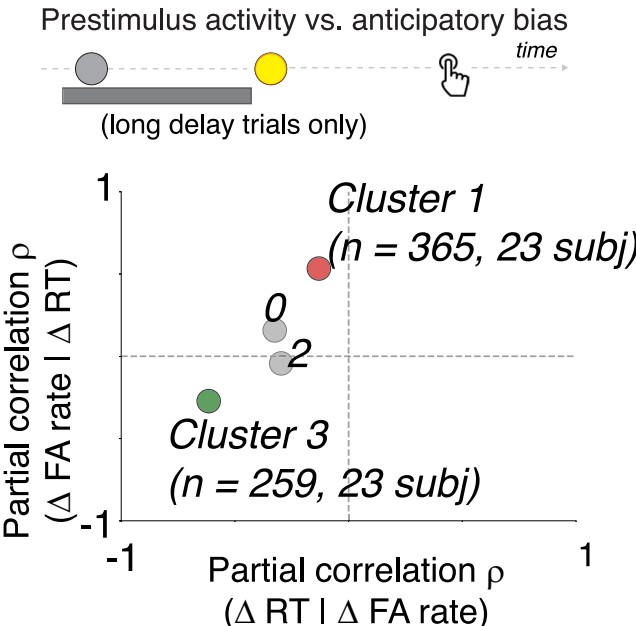

**Fig. 5 | Dissociable prestimulus neural correlates of anticipatory biases.** The top schematic shows the time interval used to compute prestimulus activity in each cluster. The scatterplot shows partial correlation coefficients between activity in each cluster and RT bias (controlling for FA bias) on the abscissa and FA bias (controlling for RT bias) on the ordinate. The green circle indicates a significant relationship with RT bias, and the red circle indicates a significant relationship with FA bias (corrected $p < 0.05$).

onset of the warning signal (Fig. 5E), presumably when participants begin preparing to respond.

The two clusters encoding distinct anticipatory biases also were associated with neural populations with distinct functional properties (Fig. 7). Cluster 1 activity patterns that encoded FA biases, on average, resembled a preparatory motor process. These activity patterns tended to rise after the warning signal, stay elevated during the foreperiod delay, and peak at the time of the motor response during all trial types (Fig. 7A–C). There also were reliable modulations by trial-to-trial RT variability, such that increased

**Fig. 6 | Neural correlates of anticipatory FA and RT biases show distinct temporal dynamics.**
**A** Time resolved partial correlation coefficient relating aggregate cluster 1 neural activity with FA bias across participants, controlling for RT bias (vertical red line indicates peak correlation, corresponding to highlighted time interval in **B**).
**B** Aggregate cluster 1 neural activity during long-delay trials across participants grouped into tertiles based on FA bias (smoothed with a Gaussian kernel). **C** Scatterplot showing across-participant correlation between FA bias and prestimulus neural activity (averaged within the 1180–1430 ms window relative to S1, as highlighted in **B**). **D** Time resolved partial correlation coefficient relating aggregate neural activity in Cluster 3 with RT bias across participants, controlling for FA bias (vertical red line indicates peak correlation, corresponding to highlighted time interval in **E**). **E** Same as (**B**) but for aggregate cluster 3 neural activity across participants grouped into tertiles based on RT bias (**F**) Scatterplot showing across-participant correlation between RT bias and prestimulus neural activity (averaged within the −120–230 ms window relative to S1, as highlighted in **E**).

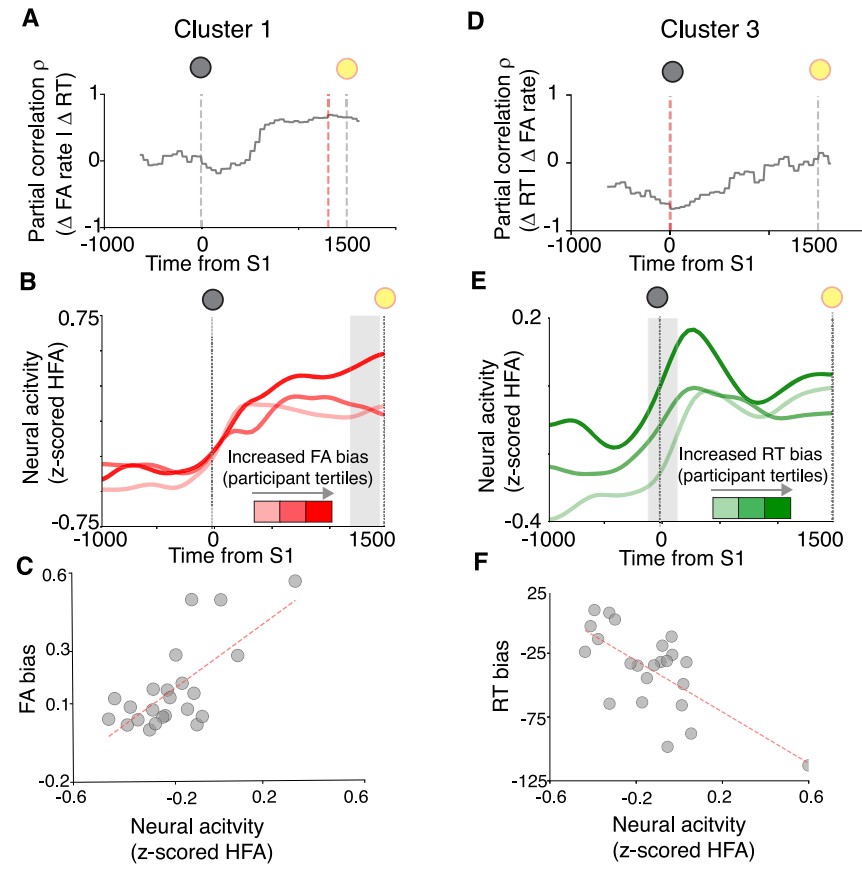

conditions of higher (more certain) anticipation. This bias is akin to pressing the car brakes sooner than necessary when you see the brake lights on the car in front of you turn on, even before the car slows down. As has been reported previously, these increased false alarms tended to be accompanied by faster RTs, thus reflecting a form of speed-accuracy trade-off[6]. We found that these FA biases were encoded by prestimulus neural activity near the end of the foreperiod delay, when the probability of the stimulus was relatively certain, in widespread neural populations that tended to ramp up just preceding the motor response and showed increased activity when RTs were faster, and were prevalent in somato-motor networks, consistent with a role in response generation[19]. Of note, these putative motor preparatory neural populations were distributed across both cerebral hemispheres even though participants always generated a motor response with the right hand. These results are consistent with prior functional neuroimaging studies that identified correlates of motor preparation in activity that spanned both cerebral hemispheres in the human brain[37,38].

The second effect, which we refer to as an RT bias, was characterized behaviorally by faster RTs under conditions of higher (more certain) anticipation. This bias is akin to pressing the car brakes faster than usual when you expect, and then see, the car in front of you slow down. This bias is consistent with well-established relationships between stimulus uncertainty and mean RT[7,35], including for very similar task designs using randomly interleaved foreperiod delays. Under these conditions, responses are thought to be suppressed while waiting the estimated duration of the short delay (when it is unknown whether it is a short- or long-delay trial) and then facilitated around the estimated end of the long delay[2,6,8,11,39]. We found that these RT biases were encoded by prestimulus neural activity at the beginning of the foreperiod delay interval, which is a reference moment for implicit time estimation, in widespread neural that tended to have transient visual responses and build gradually throughout each trial, and showed increased activity when RTs were slower, consistent with a role in visuospatial attention and response inhibition[40–42].

Previous work also identified many instances of anticipatory modulations of neural activity[2]. Our work provides new insights into those findings, leveraging the unique combination of high spatio-temporal resolution and broad anatomical sampling of iEEG measurements[30] to show that such anticipatory-driven modulations of neural activity: (1) are not limited to particular, spatially restricted sensory and/or motor neural populations, as might be inferred from animal electrophysiology studies that typically target spatially restricted recording sites[43,44]; (2) are more heterogeneous than might be inferred from scalp EEG studies that report only signals that reflect neural activity patterns that have been aggregated across large cortical areas[20–23]; and (3) have more complex temporal dynamics than might be inferred from functional neuroimaging studies with relatively low temporal resolution[24–26]. Our results also build on recent work[34] showing that that spontaneous fluctuations in neural activity that underlie intertrial variability in human behavior are far more widespread than shown by prior functional neuroimaging studies[45].

One of our key findings was that the "rise-to-bound" dynamics of our model, which have previously been used to link algorithmic descriptions of behavior to their implementations in the brain[19,31,33,45,46], had a relatively direct mapping to just a subset of the task-relevant neural signals that we identified. Instead, we found two distributed networks with task-relevant modulations of neural activity, corresponding roughly two distinct classes of algorithmic models of anticipatory behavior: (1) those that feature motor preparation[10,11,14], which encoded anticipatory biases associated with more false alarms; and (2) those that feature visuospatial attention[44,47], which encoded anticipatory biases associated with speeded responses. More generally, our results suggest that the mapping between algorithmic models fit to behavior and brain dynamics is not as straightforward as suggested by certain local and aggregate neural signals, as has been noted previously[31–33,48]. In particular, the mapping appears to involve neural signals that have diverse forms and locations, which we characterized though our clustering analysis. Results from these analyses are in-line with the emerging view that although the cortex can be segmented into distinct regions based on structural features[49,50], information processing is largely distributed across parallel, interleaved processing streams ("intrinsic

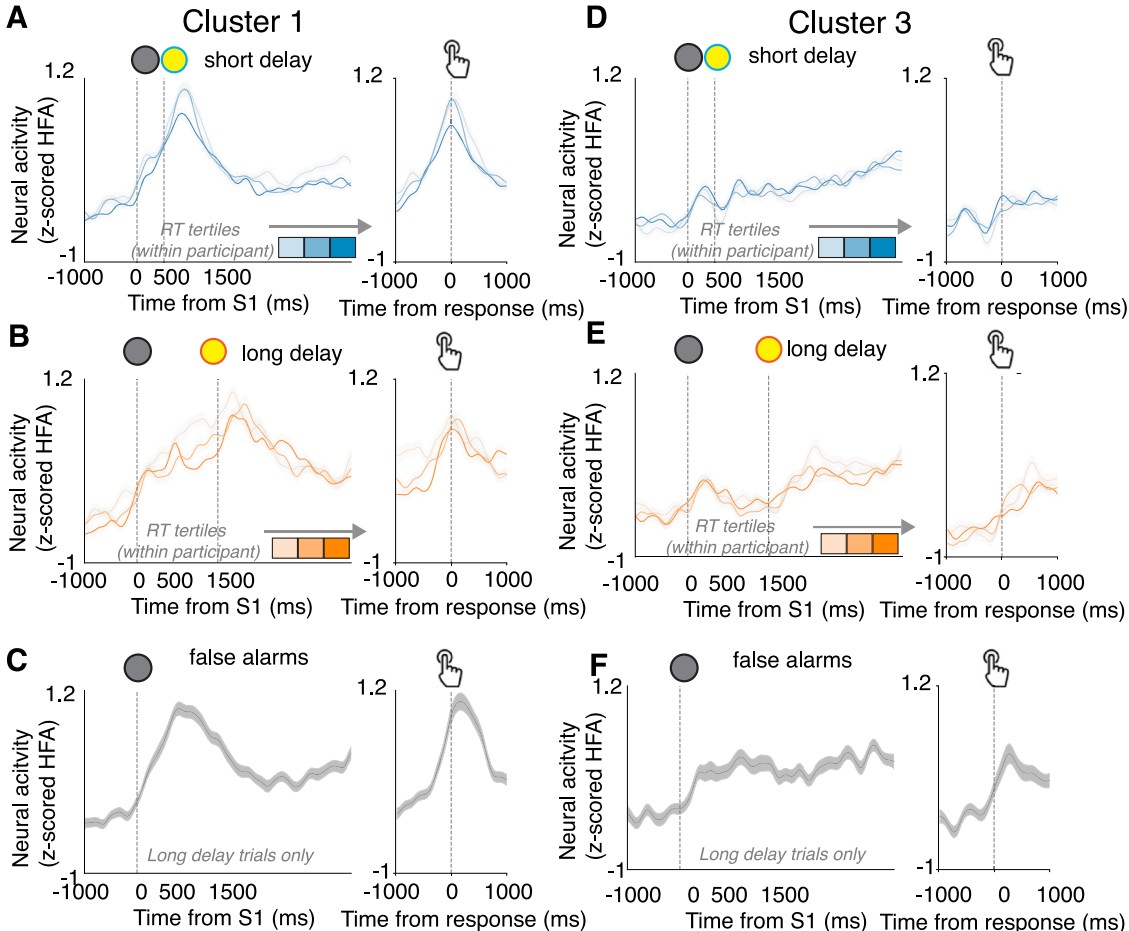

**Fig. 7 | Neural correlates of anticipatory FA and RT biases show distinct functional properties. A** Average task-related neural responses for Cluster 1 electrodes (same format as Fig. 2C, left panel) for short-delay trials associated with correct responses (RT > 250; blue). **B** Average neural activity locked to target onset (S1) for long-delay trials associated with correct responses (RT > 250; orange) (**C**) Same as A and B, but for false-alarm responses on long-delay trials (gray; S2 was not presented on these trials). **D**–**F** Same as (**A**, **B**), but for Cluster 3 electrodes.

brain networks"), with localized information processing limited to certain primary sensory and motor regions[51–53].

Our study has several limitations. First, it is possible that some of these results are particular to our patient population. In general, epilepsy patients can show additional forms of inter-individual variability in their brain networks related to their pathology[54] that might also manifest in certain categorical behavioral differences compared to healthy controls[55]. However, we sought to mitigate such concerns by focusing on neural signals associated with specific, highly controlled behaviors. Under these kinds of conditions, it has been shown that neural findings from intracranial EEG studies in patients with epilepsy can generalize to healthy controls populations[56]. Nevertheless, more work is needed to fully understand the relationship between behavioral and neural variability across individuals[57], which can have broad evolutionary[58], developmental[59], and functional[60] causes. Second, our clustering analysis was intended to identify distinct functional neural population that were evident across all subjects we studied. A more granular clustering of distinct functional profiles may be possible with a larger dataset. Third, we focused on within-trial anticipatory biases for this study but recognize that across-trial biases may also occur[11]. Further work is needed to understand if and how within- and across-trial anticipatory biases relate to each other on a neural level.

Despite these limitations, our study provided new insights into the distributed neural processes in the human brain that support anticipatory influences on sensory-motor behaviors. Despite relatively simple algorithmic explanations underlying anticipatory behavior, we found that anticipatory computations had heterogeneous, distributed, and regionally intermixed neural correlates in the human brain. These results reflect the fundamental role of anticipation in higher brain function and can help explain why simple sensory-motor processing engages widespread brain networks[61,62]. These results illustrate how high-resolution neural measurements in the human brain can complement algorithmic models in illuminating cognitive processes underlying human behavior. Moreover, these results motivate the need for more high-resolution and large-scale neural recordings combined with sophisticated computational models of brain activity and behavior to help bridge gaps in our understanding of the computational underpinnings of behavior and their complex neural substrates[60,63,64].

## Materials and methods
### Participants
We studied 23 patients with medically refractory epilepsy who underwent surgical implantation of intracranial electrodes for seizure localization (Table S2). Patients provided informed consent to perform cognitive testing as part of our research study while they were admitted to the hospital. Our study was approved by the University of Pennsylvania Institutional Review Board. Clinical circumstances alone determined the number and placement of implanted electrodes. All ethical regulations relevant to human research participants were followed.

### Stimulus-detection task
We used a stimulus-detection task with a variable foreperiod delay[6,7,36]. Participants viewed visual stimuli on a laptop computer screen and responded by

**Fig. 8 | Neural correlates of anticipatory FA and RT biases are widespread and regionally inter-mixed.** **A** Brain plot showing anatomical distribution of Cluster 1 (red) and Cluster 3 (green), which encoded FA and RT biases, respectively. **B** Percentage of electrodes in each intrinsic brain network assigned to Cluster 1 (top bars), and Cluster 3 (bottom bars). Horizontal dashed line is expected percentage assuming a uniform anatomical distribution. **C** Scatterplot showing the relative frequency of Cluster 1 electrodes (ordinate) and Cluster 3 electrodes (abscissa) in each intrinsic brain network relative to their overall (expected) frequency across the brain (z-scores; same format as Fig. 3B).

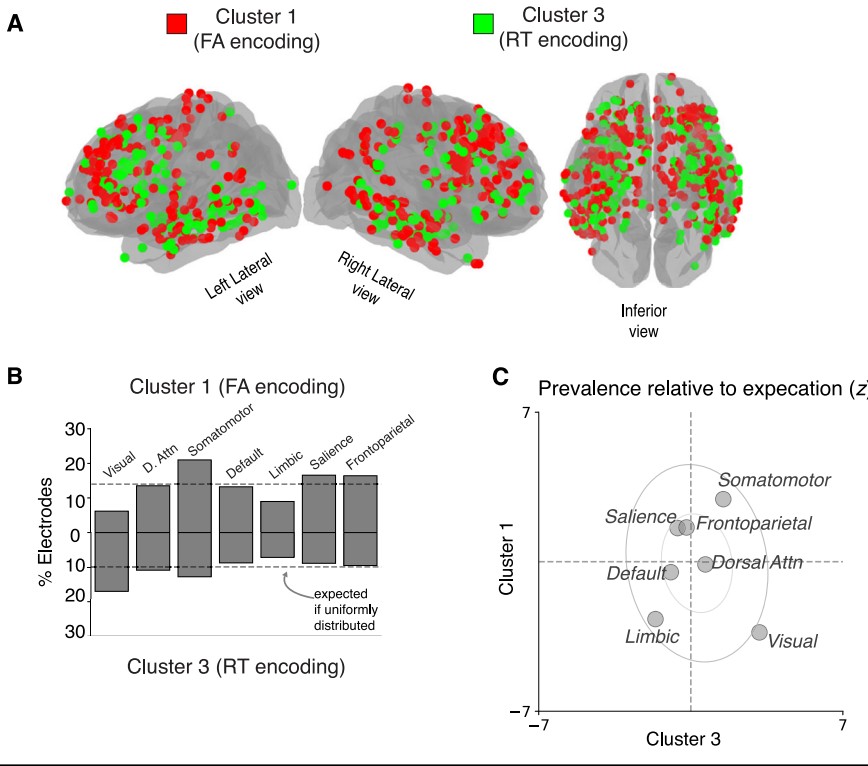

pressing a button on a game controller with their right thumb. Each trial began with the presentation of a small white box at a randomized location on the screen as a fixation target (one of nine locations on a 3 × 3 grid). The stimulus changed color to yellow after one of two randomly selected foreperiod delays: (1) short delay = 500 ms, or (2) long delay = 1500 ms (with a 50% probability of each delay condition on any given trial). We measured response time (RT) as the time between color change (stimulus) and button press (response). If a response was provided within a predefined response interval (1000 ms after color change), visual feedback was provided as follows: (1) for RTs ≤ 150 ms, salient positive feedback was shown, consisting of a smiley face with text below ("wow!"); (2) for RTs > 150 ms, the RT was shown, color-coded by binned values (green for 150–300 ms, yellow for 300–600 ms, red for 600–1000 ms). If a response occurred before the color change of the stimulus ("premature false alarm") or if no response was provided within 1000 ms after color change, the participant was shown a blank screen for 2500–2750 ms. During each session, the participant performed blocks of 9 trials each. There were no "catch" trials. After six blocks, the participant was provided with an option of performing additional blocks of trials. On average, participants performed a mean of 149 trials (i.e., 1.34 sessions). Two participants performed additional trials with a 1000 ms foreperiod delay, which we excluded from our analyses.

**Extended LATER model of RT**

All model-fitting procedures described below were implemented by finding the minimum of the appropriate objective function using the SciPy Optimization package in Python. We modeled RT distributions using an extended version the LATER model[13,14]. This rise-to-bound model describes RTs as measuring the time it takes for a latent variable to rise linearly from a starting value ($S_0$) at stimulus presentation to a threshold value ($S_t$) at response. The model accounts for trial-to-trial RT variability by assuming that the rate-of-rise of this process varies stochastically from trial-to-trial per a Gaussian process. The basic form of the LATER model describes an RT distribution using two free parameters, as follows:

$$RT \sim \frac{1}{N(\mu, \sigma^2)}$$

Where $N$ represents a Gaussian distribution with mean $\mu$ and standard deviation $\sigma$. Note that this version assumes that the threshold value $S_t = 1$; an equivalent formulation assumes that $\sigma = 1$, and $S_t$ is a free parameter. These parameters are estimated by fitting a Gaussian probability density function to reciprocal RTs, which typically resembles a Gaussian distribution[14].

We extended this basic model to account for premature false alarms and RTs < 250 ms, which we modeled as a stochastic process that occurred with uniform probability during the 500 ms preceding S2. For these and other RT analyses, we assumed that this false alarm-generating process also triggered responses within 250 ms after stimulus onset ("fast responses"), because these fast responses are thought to arise from processes distinct from those driving slower RTs (we chose 250 ms as a conservative cutoff to ensure that our neural analyses do not conflate these two processes)[6,14].

We first fit this extended LATER model to behavior on the short-delay trials. We then fit a modified model to behavior on long-delay trials that included two additional free parameters representing delay-related changes in: (1) the starting point of the LATER unit, and (2) the uniform probability of generating a premature or fast response. We evaluated the overall model fits by computing an $R^2$ value comparing the (z-scored) model-predicted probability distributions and an empirically estimated probability distribution based on a Gaussian-smoothed reciprocal RT histograms (including all responses from 500 ms prior to stimulus onset to 1000 ms following stimulus onset, with an offset such that all RTs on anticipatory false alarms were positive values and RTs on short and long-delay trials were aligned relative to stimulus onset), and also the cumulative probability of observing a premature false alarm based on the uniform distribution (see Fig. S1 for individual model fits for each participant).

**Intracranial neural recordings via intraparenchymal electrodes**

Patients were implanted exclusively with intraparenchymal depth electrodes ("stereo EEG," Ad-tech, 1.1 diameter, 4 contacts spaced 5 mm apart), except in one patient who also had subdural grid electrodes (Participant #142, Ad-tech, 4 mm contacts, spaced 10 mm apart). iEEG was recorded using a Natus recording system. Based on the amplifier and the discretion of the clinical team, signals were sampled at either 512 or 1024 Hz. Signals were converted to a bipolar montage by taking the difference of signals between

each pair of immediately adjacent contacts on the same electrode. The resulting bipolar signals were treated as new virtual electrodes (henceforth, "electrodes"), originating from the midpoint between each electrode pair[65–67]. Digital pulses synchronized the electrophysiological recordings with task events. We excluded electrodes that recorded prominent 60 Hz electrical line noise, defined as electrodes that showed greater spectral power in the 58–62 Hz range compared to the 18–22 Hz range, or electrodes that were disconnected (standard deviation = 0). We excluded trials with prominent noise artifacts (e.g., if voltage data were not recorded due to saturation, or if the mean or standard deviation of voltage was >10 standard deviations of all trials). We did not specifically exclude electrodes based on epileptic activity because our analyses focused on behaviorally linked neural activity, which should not be influenced systematically by epileptic networks[68]. We analyzed data from 2609 electrodes.

### Anatomical localization of electrodes
Intracranial electrodes were identified manually on each post-operative CT scan and contact coordinates recorded using custom software based on the center of density of the radiodense contacts on thresholded images. To obtain contact locations in each patient's native anatomic space as well as a common reference space (MNI coordinates), we used Advanced Normalization Tools[69] to register the post-operative CT to the pre-operative MRI, and the MRI to the Montreal Neurological Institute (MNI) average brain. We assigned each electrode to various canonical intrinsic brain networks ("7 network model") using a volumetric atlas in MNI coordinates[53]. We refer to the "ventral attention" network as the "salience" network based on its resemblance to behaviorally defined networks important for emotion[70], but otherwise use terminology as reported in the original study[53].

### Extracting high-frequency activity (HFA)
We extracted 5000 ms segments of iEEG data around the following task events: (1) 2000 ms prior to target onset, (2) 5000 ms after target onset, (3) the stimulus color change, and (4) onset of the motor response. For each segment, we extracted spectral power using five complex-valued Morlet wavelets (wave number 3, to increase temporal resolution) with logarithmically spaced center frequencies from 70 to 200 Hz. We squared and log-transformed the wavelet convolutions to obtain power estimates at each time sample. We removed 1000 ms buffers at the beginning and end of each segment to avoid contamination from edge artifacts. We averaged these power estimates across the 5 wavelets, resulting in a single power value for each time sample. We convolved each power-time series with a Gaussian kernel (half-width of 75 ms), resulting in a continuous representation of high-frequency activity (HFA) surrounding each task event. We z-scored HFA by the mean and standard deviation of the distribution of HFA values obtained from randomly selected segments of iEEG data recorded from that session clips (matched to the number of total task events) to not bias values towards any task-related event[65–67]. We refer to z-scored HFA as "HFA."

### Measuring and preprocessing electrode-specific activation functions
We quantified the activity pattern recorded by each electrode by measuring the average HFA around target onset during, measured separately on short-delay and long-delay trials (subsequently referred to as "activation functions"). We measured target-locked activation functions from 1000 ms prior to, until 4000 ms following target onset.

To ensure that all subsequent analyses focused on modulations of the activation functions that were separate from the immediate sensory and motor responses, we preprocessed the data as follows (all curve fits were performed using the *SciPy* Optimization package in *Python* ("cuve_fit"):

First, we estimated the overall (linear) trends in HFA that occurred over the course of the trial independent of changes that occurred immediately following stimulus presentation using the target-locked activation function (averaged across both trial types). Specifically, we fit a line to estimate how HFA changed over time using during two short segments of this activation function: (1) 1000 ms to −500 before target onset, and (2)

1000 ms after color change occurred on long-delay trial (2500 ms to 3000 ms after target onset). We subtracted these lines from each target-locked activation function to obtain detrended functions.

Second, we estimated the changes in activity that occurred immediately after target onset across both trial types. Specifically, if the detrended target-locked activation function (averaged across both trial types, from target onset to 500 ms afterwards) contained any activity peaks or troughs that were outside of the electrode's baseline activity range, we fit a Gaussian function to model these activity excursions. If both a peak and trough were detected in each electrode's activation function, we modeled whichever one was larger. We then removed the post-target Gaussian from each target-locked activation function. For long-delay trials, we also fit a Gaussian to the 500–1500 ms interval after target onset and subtracted it from the activation function to remove any expectation-related changes in activity that occurred in response to the target not changing color.

Third, we modeled activity locked to the color change separately for short- and long-delay trials. Specifically, we first fit a Gaussian to a short time segment after the color change (0–500 ms post-stimulus) to capture any changes in activity that occurred immediately after color change. We also fit any residual peaks or troughs over a longer time segment (0–1000 ms post-stimulus). We then subtracted these fit Gaussians from the activation function from each trial.

Fourth, we modeled activity locked to the motor response separately for short- and long-delay trials. Specifically, we fit separate Gaussians to the 500 ms preceding the response, to capture any pre-response ramping activity or any smeared stimulus-locked activity from the preceding color change, and to any residual peaks or troughs during the 500 ms post-response interval, to capture further changes in activity related to either the response or feedback presentation. We then subtracted these fit Gaussians from the activation function from each trial.

In summary, these procedures resulted in residual trial-by-trial HFA measures that were time-locked to four task events: (1) target-locked, short delay; (2) target-locked, long delay; (3) response-locked, short delay; and (4) response-locked, long delay.

### Relating residual neural activity to endogenous RT variability
We measured endogenous (delay-independent) RT variability for each participant as follows. First, we transformed RTs during correct responses (excluding "fast response" RTs < 250 ms) by taking the negative reciprocal ($-1/RT$), which transforms right-tailed RTs into an approximately Gaussian distribution (the negative sign is applied for convenience such that long RTs are still associated with larger values)[14]. Then, we removed delay-related RT variability by z-scoring reciprocal RTs within each delay condition.

We assessed whether endogenous RT variability was related to neural activity at a given electrode on a trial-by-trial basis, after removing stimulus- and response-locked components of the activation function, including ramping components (Fig. S3). We considered several trial-by-trial neural features locked to trial events: (1) baseline interval prior to target onset (500 ms), (2) baseline interval prior to stimulus (500 ms), (3) post-stimulus activity (250 ms), (4) post-stimulus buildup rate (slope of a line fit to HFA trend 250 ms following stimulus), (5) pre-response buildup rate (slope of a line fit to HFA 250 ms prior to response), and (6) pre-response activity (250 ms).

We performed an omnibus test to assess for any relationship between neural activity and stochastic RT variability using a multi-variate linear regression model. The dependent variable was stochastic RT variability across all trials, and the independent variables were each of the neural features described above. We measured the predictive power of this model using the sum squared error of residuals (SSE). We also assessed specific relationships between each neural feature and stochastic RT variability using Spearman correlations.

We assigned non-parametric z-statistics and p values for each of these tests by comparing each of these "true" statistics (SSE of multi-variate model, and Spearman's *rho* for each neural features) to null distributions

generated for each electrode by misaligning RTs and neural data (using a circular shift procedure, 1000 iterations, to account for any autocorrelation in RTs). For the omnibus test, we assigned a one-tailed $p$ value (i.e., if the true SSE was <5% of null SSE values, we assigned a $p$ value of 0.05). For Spearman correlations, we assigned a two-tailed $p$ value (i.e., if $rho$ was >2.5% of null $rho$ values, we assigned a $p$ value of 0.05).

## Hierarchical clustering of neural populations based on functional properties

We used data-driven unsupervised clustering to group all electrodes based on similar task-driven neural activation patterns. This approach allowed us to study task-relevant neural representations that were possibly distributed across many regions. For each electrode, we defined a multi-variate feature vector representing the magnitude of (linear) relationships between: (1) each of the five features of the activation function listed above (i.e., baseline, post-stimulus activity, post-stimulus buildup rate, pre-response buildup rate, pre-response activity) and stochastic RT variability (i.e., z-scored RT computed separately for short- and long-delay trials, thus removing any mean difference in RT for the two delay conditions), and (2) task-related modulations relative to pre-target baseline activity (Fig. 4). We characterized this feature vector as a multinomial distribution indicating the presence and direction of a significant effect ($p < 0.05$; 0 indicates no relation, or no significant task-related modulation; 1 indicates relatively increased activity with long RTs or task-related increases in activity; and -1 indicates relatively increased activity with short RTs or task-related decreases in activity). We used an agglomerative clustering algorithm (*scikit learn, Python*) to group all electrodes using Euclidean distance as a measure of pairwise similarity and a linkage function that merges clusters to minimize the variance within all clusters. We identified 4 clusters that were grouped by similar functional profiles using an objective function maximized the number of clusters that were observed in all participants and contained at least 200 electrodes.

## Statistics and reproducibility

We performed non-parametric statistical tests as described above, when appropriate. Otherwise, we performed $t$-tests to compare continuous distributions and binomial to compare categorical distributions (counts data). We used partial correlation analysis to assess across-participant correlations between neural activity and anticipatory biases (delay-related differences in mean RT and false alarm rate). Based on our partial correlation analysis, we specifically assessed for a functional dissociation between prestimulus activity modulations in Cluster 1 and Cluster 3 using a linear mixed-effects model, as follows:

$$Y \sim 1 + (\Delta\mathrm{RT} \times time \times cluster) + (\Delta\mathrm{FA} \times time \times cluster) + (1| + subj) + (1|roi)$$

Where Y is a continuous variable that represents neural activity (z-scored HFA) averaged within each subject for a specific cluster (Cluster 1 or 3), and a specific time window (−120–130 ms, "early" or 1180–1430 ms, "late" relative to S1, as identified by the partial correlation analyses, Figs. 5, 6). The Fixed Effects are two separate three-way interactions between ΔRT (continuous variable indicating delay-related difference in mean RT for each participant), time (categorical variable indicating early or late time window), and cluster (categorical variable indicating Cluster 1 or 3), and between ΔFA (continuous variable indicating delay-related difference in FA rate for each participant), time, and cluster. The Random Effects are subject ("subj") and intrinsic brain network ("roi").

We used False Discovery Rate (FDR) correction for multiple comparisons (Benjamini & Hochberg, 1995) and considered an FDR-corrected $p < 0.05$ to be statistically significant. We report FDR-corrected $p$ values when indicated, otherwise reported $p$ values are uncorrected. We performed most of our analyses using Python using both custom code and publicly available packages (e.g., *NumPy* for numerical computing, *SciPy* for statistics and signal processing, *MNE* for spectral analyses, *statsmodels* for regression modeling, *pingouin* for partial correlation analysis). We fit linear mixed effects models using in R using the lme4 and lmeTest packages[71,72].

## Reporting summary

Further information on research design is available in the Nature Portfolio Reporting Summary linked to this article.

## Data availability

The raw data used for this study are publicly available on OpenNeuro.org (https://doi.org/10.18112/openneuro.ds005624.v1.0.0). The raw and processed data that were analyzed in this study can also be obtained by contacting the corresponding author.

## Code availability

The Python code used to process these data are available online (https://github.com/ashwinramayya/code_RamaEtal_AntReact).

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

## Acknowledgements

We thank Drs. Michael Kahana, Michael Beauchamp, Tahra Eissa, Matthew Nassar, Long Ding, Kareem Zaghloul, Daniel Yoshor, Alex Vaz, and William Songjun Li for helpful feedback on the manuscript. We thank Cameron Brandon, Everett Prince, John Bernabei, Jacqueline Boccanfuso, and Joel Stein for technical assistance in data collection. This research was supported by National Institutes of Health grant 6T32NS091006 (AGR).

## Author contributions

A.G.R.: Conceptualization, Methodology, Software, Formal Analysis, Visualization, Writing – Original Draft; V.B.: Conceptualization, Investigation, Data Curation, Writing – Reviewing and Editing, Project Administration; A.R.: Investigation, Writing – Reviewing and Editing, Project Administration; T.L.: Supervision, Resources, Writing – Reviewing and Editing, Funding Acquisition; J.G.: Supervision, Conceptualization, Funding Acquisition, Writing – Original Draft.

## Competing interests

The authors declare no competing interests.
