## [Transparent Peer Review file · Communications Biology]

Human Response Times are Governed by Dual Anticipatory Processes with Distinct Neural Signatures

Corresponding Author: Dr Ashwin Ramayya

Version 0:

Reviewer comments:

Reviewer #1

(Remarks to the Author)

In this manuscript, Ramayya et al. studied 23 patients at UPenn Hospital and collected intracranial EEG data during a classic stimulus detection task with a variable foreperiod delay. They confirmed that the behavioral results of the patients were consistent with previous observations in normal healthy individuals performing the same task (i.e. faster RT and higher false alarm on long vs. short delay trials). Interestingly, these behavioral signatures were demonstrated to have functionally (but not anatomically according to the authors) dissociable neural correlates: increased incorrect/premature/false alarms were associated with increased broadband gamma activity (BGA) prior to response-cue (S2) whereas faster RT during correct long delay trials were associated with increased BGA following S1. Overall, the work is presented nicely and has a good methodological rigor. Here are some comments that may help the authors improve the quality of their work further, in particular to better use previous fMRI results to explore smaller regions of interest in their data-set. (see below).

1) The modeling approach: I understand the idea to a priori postulate that the starting point parameter would better capture the behavioral effects given the design of the study. Would't the results reinforced if you could demonstrate this more formally ? For instance by performing a model comparison test to eliminate other plausible implementations/parameters of rise-to bound models that were tested (and presented latter on page 9 of the results). Perhaps a Bayesian model comparison could demonstrate that the starting point better capture behavioral variability ? Model simulations vs. data at the group level would probably help (it is nice to show individuals as in Figures S1., but it would be interesting to see how different models predict the core behavioral findings (i.e. how the different models capture the effects reported in fig. 1 C-D, showing all individuals as dots + adding the average model predictions with SEM to such a figure would be informative)

2) Along the same line, the conceptual justification to use two sets of starting point parameters to account for the behavioral results is not explained (in the ms. This is justified by " a large inter-subject variability" p.8, line 127-28); Would it be possible to show the model's simulations vs. behavioral data to compare 1 vs. 2 parameters predictions ? Since it would be more parsimonious to use a single parameter to account for both types of effect if this similarly account for the two behavioral effects of interest.

3) I found the example electrode of Fig. 2C a little challenging to follow partly because showing more than ten curves within a panel might not be optimal; I suggest to re-work on this part of the manuscript, perhaps either by (i) using tertiles (3 curves) or quartiles (4 curves) per condition + consider separating short vs. long trials in separate plots ? Alternatively, the authors could show single trials (raster plot-like BGA), separately for short and long delay trials and order the single trials as a function of RT (this would be more easy to see the effects: the reader would see that BGA peaks at response (that would be illustrated by a black dot for example for each trial; and the amplitude of BGA for a given trial could be color coded such that each row would correspond to a trial); Also since after all the core effects of interest are derived from long delay trials, perhaps focusing on these and how neural activity predicts on two single electrodes the RT bias or the FA effect would be more coherent with the rest of the manuscript ?

4) The examination of 'RT modulations' (which do not depend on the process studied in this task) could be used as a control to show that task-related effects are specific); indeed, although interesting, I found these task-independent results rather distracting and confusing to the main message of the paper; I suggest presenting it in the supplementary material rather than in the main figures because it appears as a non-specific behavioural marker, which almost mechanically leads to finding non-specific neural associations. This would allow the paper to better focus on the two anticipatory processes and their

anatomical neural correlates (see my next point).

5) Anatomical approach: The justification for using such large regions of interest is unclear; Given the fact that this is a relatively old task (1965), numerous studies were done using non-invasive neuroimaging methods which findings could be used to investigate the possibility that more specific neural correlates might actually map to the findings reported in figs 6-7; To be more specific, I suggest to complement the findings using a finer grained parcellation scheme (for example AAL parcels or Destrieux/Free surfer parcellation scheme or any alternative that split the brain more finely than across the 7 "networks used in this study"); This would allow a better comparison with prior fMRI work and with prior monkey's single cell evidences (which could be better integrated in the ms.); For example, I quickly checked some of the paper the authors cite using fMRI: Cui et al. 2009 findings suggest that SMA and STG neural activities should differ between short vs. long delay trials; using iEEG to decipher the neural dynamics of such more specific structures would therefore be highly interesting. Hence, my point is that using such very large parcels in this study somehow guarantees to find that "widely distributed" or "uniformally distributed" neural activations associated with the cognitive processes of interest. Would it be possible to better use previous fMRI findings to delineate finer regions of interest (such as dorsal vs. ventral anterior insula vs. dorsal vs. midd. cingulate, SMA vs. pre-SMA ? IFG vs. dIPFC ? since dIPFC was indeed studied in Vallesi et al., 2009 ?). Because it is so difficult and time-consuming to acquire iEEG data, pooling a lot of site within very large parcels appears suboptimal (although the authors did a great job at summarizing the core BGA pattern corresponding to the behavioral signatures of interest using such a global approach). Yet, complementing their findings with a more fine-grained data-analysis pipeline would better capitalize on the very good spatio-temporal resolution of intracranial EEG; I thus encourage the authors to consider to use a finer brain parcellation scheme while relaxing in the same time the methodological constraints they imposed on their data (for instance at least 5 participants showing a significant effect within a parcel + at least 10 significant iEEG sites within a smaller parcel would nicely complement their initial findings ?). In sum, it would be very interesting to complement fig. 6 and fig 7 with finer grained- anatomical information using smaller regions of interests.

Minor:

-Cui et al. Plos Biol 2009 is missing in the reference list

- I found that the use of the term "false alarm" to label the trials in several instances was perhaps not optimal: for instance, in the two core figures of the paper (figs 6-7, panel B: in orange I would label this more simply as correct long delay trials while in grey these are ALSO long delay trials but incorrect, right ?); Perhaps just sticking to correct vs. incorrect long delays would simplify the paper ?

At the end, I congratulate the authors for a very interesting work!

Reviewer #2

(Remarks to the Author)

This is an interesting study showing two distinct types of anticipatory neural activity in a motor preparation task. In a model-based analysis, the authors show that false alarms and faster reaction times to expected targets are preceded by neural responses in preparatory motor and visuospatial activity respectively. One noteworthy aspect of their work is the high number of participants with implanted electrodes. I believe this is a solid study, and will be of great interest to the temporal expectation and motor preparation literature. I have only a few minor comments that would require clarification.

The task description is missing some important information. I'm assuming that short and long trials had the same probability (50/50), but this is not clearly stated in the paper. Similarly, I presume that there were no catch trials, but it needs to be included in the methods.

The authors correctly suggest that the effect of cluster 1 resembles motor preparatory activity, but I'm curious to know why they think this activity was not lateralized, since subjects always responded with their right thumb. The lack of any clear lateralization might be worth including in the discussion.

Figure 6B has no axes labels.

Version 1:

Reviewer comments:

Reviewer #1

(Remarks to the Author)

The corrections made have significantly improved the manuscript and the authors' responses are satisfactory. So I have no further comments.

Congratulations to the authors for this great revision

Reviewer 1 Comments	Authors' Response
In this manuscript, Ramayya et al. studied 23 patients at UPenn Hospital and collected intracranial EEG data during a classic stimulus detection task with a variable foreperiod delay. They confirmed that the behavioral results of the patients were consistent with previous observations in normal healthy individuals performing the same task (i.e. faster RT and higher false alarm on long vs. short delay trials). Interestingly, these behavioral signatures were demonstrated to have functionally (but not anatomically according to the authors) dissociable neural correlates: increased incorrect/premature/false alarms were associated with increased broadband gamma activity (BGA) prior to response-cue (S2) whereas faster RT during correct long delay trials were associated with increased BGA following S1. Overall, the work is presented nicely and has a good methodological rigor. Here are some comments that may help the authors improve the quality of their work further, in particular to better use previous fMRI results to explore smaller regions of interest in their data-set. (see below).	We appreciate the reviewer's excellent suggestions for improving our manuscript and recognition of our work.
1) The modeling approach: I understand the idea a priori postulate that the starting point parameter would better capture the behavioral effects given the design of the study. Would't the results reinforced if you could demonstrate this more formally? For instance by performing a model comparison test to eliminate other plausible implementations/parameters of rise-to bound models that were tested (and presented latter on page 9 of the results). Perhaps a Bayesian model comparison could	We clarified and expanded the text to describing the results of a formal model-comparison analysis that was included in the original manuscript (Lines 145–159). Specifically, we formally compared the fits of three models of anticipatory modulation of RT distributions: 1) changes in starting point, 2) mean rate of rise, and 3) variance of rate of rise. Each of these models contained the same number of free parameters, such that we could directly compare goodness-of-fit (using R^2) without the need to apply a penalty for model complexity. In summary, model 3 produced poorer fits of the data, but model 1 and model 2 produced similar fits. We focused on model 1 for the remainder of the manuscript because it provides a more parsimonious explanation for the neural data.

demonstrate that the starting point better capture behavioral variability ?

Model simulations vs. data at the group level would probably help (it is nice to show individuals as in Figures S1., but it would be interesting to see how different models predict the core behavioral findings (i.e. how the different models capture the effects reported in fig. 1 C-D, showing all individuals as dots + adding the average model predictions with SEM to such a figure would be informative)

2) Along the same line, the conceptual justification to use two sets of starting point parameters to account for the behavioral results is not explained (in the ms. This is justified by “ a large inter-subject variability” p.8, line 127-28); Would it be possible to show the model’s simulations vs. behavioral data to compare 1 vs. 2 parameters predictions ? Since it would be more parsimonious to use a single parameter to account for both types of effect if this similarly account for the two behavioral effects of interest.

We appreciate this excellent suggestion. We performed new model simulation analyses and present the results in the new Figure S2. These simulations further support the formal model comparisons, showing that models 1 and 2 are roughly equally consistent with the data and provide better accounts than model 3. We also used these simulations to justify our approach of using independent modulations of prestimulus and poststimulus starting points when modeling anticipatory bias (Supplemental Text; Fig. S2). The new Figure S2 is reproduced below.

Figure S2. Model Simulations. (A) Example RT distribution from short-delay trials from a human participant (reproduced from Fig. 1A). (B) Simulated RT distribution from the base model of RT behavior on short-delay trials. (C) Schematic illustration of the poststimulus rise-to-bound process underlying RT distributions. (D-I) Model simulations of anticipatory FA bias (ordinate) and RT bias (abscissa). The black box indicates the range of observed group-level changes in mean RT and FA biases observed in human participants (as shown in Fig. 1E). Each gray circle represents simulated behavioral data from a unique parametric modification to the base model, as follows: (D) independent changes in prestimulus starting point and poststimulus starting point modulations (Model 1); (E) independent changes in prestimulus starting point and mean of poststimulus rate-of-rise (Model 2); (F) independent changes in prestimulus starting point and variance of poststimulus rate-of-rise (Model 3); (G) Isolated changes in prestimulus starting point; (H)

Isolated changes in poststimulus starting point; (I) Simultaneous (correlated) changes prestimulus and poststimulus starting points.

3) I found the example electrode of Fig. 2C a little challenging to follow partly because showing more than ten curves within a panel might not be optimal; I suggest to re-work on this part of the manuscript, perhaps either by (i) using tertiles (3 curves) or quartiles (4 curves) per condition + consider separating short vs. long trials in separate plots ? Alternatively, the authors could show single trials (raster plot –like BGA), separately for short and long delay trials and order the single trials as a function of RT (this would be more easy to see the effects: the reader would see that BGA peaks at response (that would be illustrated by a black dot for example for each trial; and the amplitude of BGA for a given trial could be color coded such that each row would correspond to a trial); Also since after all the core effects of interest are derived from long delay trials, perhaps focusing on these and how neural activity predicts on two single electrodes

We followed the reviewer’s first suggestion of reproducing the figure using tertiles of RT and by separating short and long delay conditions. The revised figure is shown below. We also revised the new Fig. 7 (previously Figs. 6A and 7A) to use the suggested format to represent aggregate neural activity for Clusters 1 and 3.

We also revised Figure 7 to use the suggested format to represent aggregate neural activity for Cluster 1 and Cluster 3 (Fig. 7, shown below; previously Figs. 6A and 7A).

the RT bias or the FA effect would be more coherent with the rest of the manuscript ?

4) The examination of 'RT modulations' (which do not depend on the process studied in this task) could be used as a control to show that task-related effects are specific); indeed, although interesting, I found these task-independent results rather distracting and confusing to the main message of the paper; I suggest presenting it in the supplementary material rather than in the main figures because it appears as a non-specific behavioural marker, which almost mechanically leads to finding non-specific neural associations. This would allow the paper to better focus on the two anticipatory processes and their anatomical neural correlates (see my next point).

We regret the unfortunate wording we used in several places that seemed to imply that RT modulations are “task independent.” We have revised the text to clarify that neural activity that reflects the trial-by-trial variations in RT are a useful marker of activity that is relevant to performance on the task, although these modulations occur independently of the delay-related manipulations we used in our task. Accordingly, to identify the two anticipatory processes that we describe, we first needed to identify candidate recording locations based on these RT modulations. Moreover, the nature and timing of the RT modulations allowed us to make inferences about the functional roles of neurons these locations. In particular, the cluster that showed negative trial-by-trial correlations with RTs (increased activity corresponded to faster RTs; Cluster 1 in Fig. 4) was consistent with a role in motor preparation. Neurons in this cluster had prestimulus activity patterns that encoded the propensity of participants to generate false alarms on long delay trials, but not the propensity to generate speeded responses (Fig. 5; Fig. 6A–C). In contrast, the cluster that showed positive correlations with RTs (increased activity corresponded to slower RTs; Cluster 3 in Fig. 4) was more consistent with a role in attention and response inhibition. Neurons in this cluster encoded the propensity of participants to generate speeded responses on long delay trials, but not the propensity to generate false alarms responses (Fig. 5; Fig. 6D–F). We clarify these points throughout the manuscript.

5) Anatomical approach: The justification for using such large regions of interest is unclear; Given the fact that this is a relatively old task (1965), numerous studies were done using non-invasive neuroimaging methods which findings could be used to investigate the possibility

We performed additional analyses using finer grained-anatomical localization (Figure S4). Overall, our findings were similar to those we presented with intrinsic brain networks. These results further motivate our approach of assessing for correlates of anticipatory biases across clusters of functionally defined neural populations, rather than specific brain regions (see Supplemental Text; Fig. S4). The new Figure S4 is reproduced below.

that more specific neural correlates might actually map to the findings reported in figs 6-7; To be more specific, I suggest to complement the findings using a finer grained parcellation scheme (for example AAL parcels or Destrieux/Free surfer parcellation scheme or any alternative that split the brain more finely than across the 7 “networks used in this study”); This would allow a better comparison with prior fMRI work and with prior monkey’s single cell evidences (which could be better integrated in the ms.); For example, I quickly checked some of the paper the authors cite using fMRI: Cui et al. 2009 findings suggest that SMA and STG neural activities should differ between short vs. long delay trials; using iEEG to decipher the neural dynamics of such more specific structures would therefore be highly interesting. Hence, my point is that using such very large parcels in this study somehow guarantees to find that “widely distributed” or “uniformly distributed” neural activations associated with the cognitive processes of interest. Would it be possible to better use previous fMRI findings to delineate finer regions of interest (such as dorsal vs. ventral anterior insula vs. dorsal vs. mid. cingulate, SMA vs. pre-SMA ? IFG vs. dIPFC ? since dIPFC was indeed studied in Vallesi et al., 2009 ?). Because it is so difficult and time-consuming to acquire iEEG data, pooling a lot of site within very large parcels appears suboptimal (although the authors did a great job at summarizing the core BGA pattern corresponding to the behavioral signatures of interest using such a global approach). Yet, complementing their findings with a more fine-grained data-analysis pipeline would better capitalize on

Figure S4. Anatomical parcellation with brain regions. (A) Brain plot showing electrode locations from all participants in standard MNI coordinates. Colors indicate brain regions of interest based on co-registration with normative atlases (Table S1). (B) Scatterplot showing the relative frequency of electrodes with positive (ordinate) and negative (abscissa) changes in activity to task-relevant events in each intrinsic brain network relative to their overall (expected) frequency across the brain (z-scores). Positive values indicate increased relative frequency; negative values indicate decreased relative frequency. Inner and outer ellipses indicate 1σ and 2σ confidence intervals derived from the joint distribution, respectively. (C) Same as B, but for positive and negative correlations with delay-independent, trial-to-trial RT variability during any time interval. (D) The top schematic shows the time interval used to compute prestimulus activity in each region of interest. The scatterplot shows partial correlation coefficients for RT bias (controlling for FA bias) on the abscissa and FA bias (controlling for RT bias) on the ordinate. (E) Same as D, but for specific regions implicated by prior functional neuroimaging studies (Cui et al. 2009; Vallesi et al 2009).

the very good spatio-temporal resolution of intracranial EEG; I thus encourage the authors to consider to use a finer brain parcellation scheme while relaxing in the same time the methodological constraints they imposed on their data (for instance at least 5 participants showing a significant effect within a parcel + at least 10 significant iEEG sites within a smaller parcel would nicely complement their initial findings ?). In sum, it would be very interesting to complement fig. 6 and fig 7 with finer grained-anatomical information using smaller regions of interests.	
Minor: -Cui et al. Plos Biol 2009 is missing in the reference list	We have added this reference in the list (Line 930) Cui, X., Stetson, C., Montague, P. R., & Eagleman, D. M. (2009). Ready... go: amplitude of the fMRI signal encodes expectation of cue arrival time. PLoS biology, 7(8), e1000167.
- I found that the use of the term “false alarm” to label the trials in several instances was perhaps not optimal: for instance, in the two core figures of the paper (figs 6-7, panel B: in orange I would label this more simply as correct long delay trials while in grey these are ALSO long delay trials but incorrect, right ?); Perhaps just sticking to correct vs. incorrect long delays would simplify the paper ?	We appreciate the suggestion but decided to keep the “false alarm” terminology because it is more specific than “incorrect.” For this task, “incorrect” trials can also occur when the participant does not generate a response within the 1 second poststimulus interval. We have revised Figures 6 and 7 for clarity, including better and more consistent labels. The updated figures are shown below Figure 6:

Figure 7:

At the end, I congratulate the authors for a very interesting work!	We appreciate the tremendously insightful comments that have greatly improved our manuscript.
Reviewer 2	Authors' Response
This is an interesting study showing two distinct types of anticipatory neural activity in a motor preparation task. In a model-based analysis, the authors show that false alarms and faster reaction times to expected targets are preceded by neural responses in preparatory motor and visuospatial activity respectively. One noteworthy aspect of their work is the high number of participants with implanted electrodes. I believe this is a solid study, and will be of great interest to the temporal expectation and motor preparation literature. I have only a few minor comments that would require clarification.	We appreciate the reviewer's comments and recognition of our work.
The task description is missing some important information. I'm assuming that short and long trials had the same probability (50/50), but this is not clearly stated in the paper. Similarly, I presume that there were no catch trials, but it needs to be included in the methods.	We have included these details in the description of the methods. The reviewer is correct, we have clarified these points in the manuscript (Lines 655 and 664). The relevant text is reproduced below (with changes italicized): Line 655: The stimulus changed color to yellow after one of two randomly selected foreperiod delays: 1) short delay=500 ms, or 2) long delay=1500 ms (with a 50% probability of each delay condition on any given trial). Line 664: During each session, the participant performed blocks of 9 trials each. There were no "catch" trials.
The authors correctly suggest that the effect of cluster 1 resembles motor preparatory activity, but I'm curious to know why they think this activity was not lateralized, since subjects always responded with their right thumb. The lack of any clear lateralization might be worth including in the discussion.	We have expanded the discussion section in response to this point (line 548-552). The updated text is reproduced below: Of note, these putative motor preparatory neural populations were distributed across both cerebral hemispheres even though participants always generated a motor response with the right hand. These results are consistent with prior

functional neuroimaging studies that identified correlates of motor preparation in activity that spanned both cerebral hemispheres in the human brain (Adam et al 2003; Hnakawa et al 2008).

Additional References

Adam, J. J., Backes, W., Rijcken, J., Hofman, P., Kuipers, H., & Jolles, J. (2003). Rapid visuomotor preparation in the human brain: a functional MRI study. *Cognitive Brain Research*, 16(1), 1-10.

Hanakawa, T., Dimyan, M. A., & Hallett, M. (2008). Motor planning, imagery, and execution in the distributed motor network: a time-course study with functional MRI. *Cerebral cortex*, 18(12), 2775-2788.

Figure 6B has no axes labels.

We have updated Figures 6 and 7 to improve clarity, including new axes labels throughout. The updated figures are shown below:

Figure 6:

Figure 7: